# Topological Zigzag Spaghetti for Diffusion-based Generation and Prediction on Graphs

**Yuzhou Chen**[1]  **Yulia R. Gel**[2]
[1]Department of Statistics, University of California, Riverside
[2]Department of Statistics, Virginia Tech
`yuzhou.chen@ucr.edu`
`ygl@vt.edu`

## Abstract

Diffusion models have recently emerged as a new powerful machinery for generative artificial intelligence on graphs, with applications ranging from drug design to knowledge discovery. However, despite their high potential, most, if not all, existing graph diffusion models are limited in their ability to holistically describe the intrinsic *higher-order* topological graph properties, which obstructs model generalizability and adoption for downstream tasks. We address this fundamental challenge and extract the latent salient topological graph descriptors at different resolutions by leveraging zigzag persistence. We develop a new computationally efficient topological summary, zigzag spaghetti (ZS), which delivers the most inherent topological properties *simultaneously over a sequence of graphs at multiple resolutions*. We derive theoretical stability guarantees of ZS and present the first attempt to integrate dynamic topological information into graph diffusion models. Our extensive experiments on graph classification and prediction tasks suggest that ZS has a high promise not only to enhance performance of graph diffusion models, with gains up 10%, but also to substantially booster model robustness.

## 1 Introduction

Diffusion models on graphs, as a novel generative paradigm in artificial intelligence, with applications from drug design to material discovery, has invigorated interest of the deep learning (DL) community in developing more systematic, efficient, and reliable mechanisms for graph representation learning (Yang et al., 2023; Zhang et al., 2023; Liu et al., 2024). In turn, the emergence of probabilistic diffusion models has sparked attention to diffusion-based graph generation, e.g., decomposition of the full diffusion into multiple simpler inter-related diffusion processes to model the dependencies among nodes and edges (Jo et al., 2022), which demands diffusion models to accurately capture intrinsic higher-order topological properties simultaneously across multiple (sub)graphs. However, prevailing graph diffusion models tend to exhibit limited abilities to describe such key joint topological characteristics across multiple objects, which obstructs their generalizability and restricts their utility for downstream tasks (Kong et al., 2023; Yi et al., 2024a; Cai et al., 2024). Addressing this barrier requires development of new mathematical approaches, enabling us to simultaneously extract the most illustrative topological characteristics *not of a single graph* but of *a sequence of graphs*, with the problem being further exacerbated for diffusion models for time-evolving graphs.

We postulate that this fundamental challenge can be approached by blending the emerging ideas of graph diffusion models with the mathematical machinery of zigzag persistence (ZP). **What is the premise?** Persistent homology (PH) on graphs allows us to learn the key higher order shape descriptors, i.e., properties that are, broadly speaking, invariant under continuous transformations such as twisting, compressing, and stretching (Carlsson, 2009). Combination of PH with DL on graphs, usually in a form of a fully trainable topological layer, often results in superior graph learning performance and higher robustness to perturbations (for the recent overviews see, e.g., Yan et al. (2021); Horn et al. (2022); Verma et al. (2024)). However, traditional PH focuses *only on a single graph*. In contrast, ZP is the powerful mathematical tool based on the theory of quiver representations that allows us to advance the ideas of the traditional PH to a case of simultaneous evaluation of the key shape characteristics of *a sequence of graphs* (Carlsson & Silva, 2010; Tausz & Carlsson,

2011). The extracted zigzag topological information can be then conveniently summarized in a form of zigzag persistence image or zigzag filtration curves (Chen et al., 2021; 2022). Such summaries satisfy the conditions of Lipschitz continuity and, as such, are suitable as input to a fully trainable topological layers in DL on par with the traditional PH tools. However, these ZP summaries require some a-priori knowledge of the data and yield topological information extracted *only* for a single user-predefined resolution scale. To mitigate this problem, Xian et al. (2022) developed a crocker plot. Crocker plot does not use the ZP notion per say, but is based on the traditional PH framework, recording the number of topological features at each resolution. Although being tractable and computationally efficient, crocker plots are not differentiable and cannot serve as an input to a fully trainable topological layer. Furthermore, crocker plots yield only local information on the graph topology, bypassing critical information on lifespans of topological features. These open questions on zigzag topological summaries, along with computational costs of ZP on graphs have been obstructing broader applicability of ZP and keeping it largely as a theoretical concept in algebraic topology, albeit a number of recent studies demonstrating the ZP potential in ecology, engineering, and social sciences (Mata et al., 2015; Myers et al., 2023b; McDonald et al., 2023; Chen et al., 2023) and albeit the recent advances in improving computational efficiency of ZP on graphs (Dey & Hou, 2021; Dey et al., 2023).

Here we propose to bridge the rising research directions on graph diffusion representations and zigzag persistence. We develop a new computationally efficient time-aware topological summary, *zigzag spaghetti* (ZS), which simultaneously captures the key joint higher-order shape properties from a sequence of graphs at all resolution scales. We show that ZS enjoys the important theoretical stability guarantees which in practice imply resistance of ZS to uncertainties and, hence, is of particular significance for diffusion-based graph generation tasks, often involving noisy conditions and limited data scenarios. We also explore the applicability of ZP as a backbone tool for topological bootstrap and the associated topological uncertainty quantification, resulting in reducing variability up to 5 times comparing to the competing models. Finally, we validate the ZS utility in diffusion-based prediction and classification tasks on graphs, illustrating the critical role the latent higher order topological information plays in performance and robustness of graph diffusion models.

Significance of our contributions can be summarized as follows:

- To the best of our knowledge, this is the first attempt to bridge not only zigzag persistence but generally, tools from algebraic and computational topology with generative diffusion models on graphs.
- We develop a new computationally efficient and easily tractable time-aware topological summary, *zigzag spaghetti*, for simultaneous assessment of the latent topological characteristics of multiple graphs at various resolution scales and derive its theoretical stability guarantees.
- We show the utility of ZS in application to extracting time-conditioned topological knowledge from time-evolving graphs and to topological uncertainty quantification.
- With extensive experiments on a broad range of benchmark datasets, we demonstrate the superiority of our ZS-based tools over the strongest state-of-the-art graph-based models for both graph prediction and classification tasks, resulting not only in performance gains up to 10%, but prominent improvement in robustness.

## 2 RELATED WORK

**Diffusion Models** Inspired by non-equilibrium thermodynamics, diffusion models are proposed as a tool to reconstruct data samples from noise by reversing the diffusion process of the Markov Chains (Sohl-Dickstein et al., 2015; Yang et al., 2023). Recently, diffusion-based tools have sprung up as a new branch of generative models such as Stochastic Differential Equations (SDE) (Song et al., 2020c), Denoising Diffusion Probabilistic Models (DDPM) (Ho et al., 2020), Denoising Diffusion Implicit Models (DDIM) (Song et al., 2020b), Noise Conditional Score Networks (NCSN) (Song & Ermon, 2019), and autoregressive diffusion model for graph generation (Kong et al., 2023). Additionally, DiGress of Vignac et al. (2023) consider a discrete diffusion process to progressively add discrete noise to graphs by either creating or removing edges and altering node categories for graph generation, while LGD of Cai et al. (2024) applies a score-based diffusion generative model in the latent space to generate new graph representations. However, most graph diffusion approaches

tend to overlook the richer set of topological and structural information in the input data. To address this barrier, TopoGAN model with a generative adversarial network (GAN) framework is used to bridge synthetic and real data distributions in the topological feature space (Wang et al., 2020), while Niu et al. (2020) develop a permutation invariant model to study the gradient of the distribution of input data. Nevertheless, these approaches still tend to suffer from the following limitations. First, they can only model spatial information, leading to the restricted capabilities of capturing (long-term) inter-dependencies and intra-dependencies. Second, they tend to be limited in quantifying uncertainties, which hampers their applicability under scenarios with scarce records and unseen data.

**Zigzag Persistence** has recently emerged as a new powerful machinery in computational topology. ZP has proven its utility in a wide range of domains, particularly, involving time-evolving and dynamic objects, such as neuronal images and brain functions (Mata et al., 2015; Chowdhury et al., 2018), swarming phenomena in biology (Corcoran & Jones, 2017; Kim et al., 2020), cryptocurrency analytics (Chen et al., 2021; 2022), commuting trends in transportation networks (Myers et al., 2023b), coral reef resilience (McDonald et al., 2023), power distribution planing (Chen et al., 2023), and cybersecurity (Myers et al., 2023a). Nevertheless, beyond a handful of recent studies (Chen et al., 2021; 2022), the ZP utility in DL still remains largely unexplored (Carlsson & Gabrielsson, 2020). One of the primary roadblocks on the way of ZP on graphs (arguably) remains computational costs, although recently there has been a notable progress in this direction (Dey et al., 2014; Dey & Hou, 2021; Dey et al., 2023). In addition, the existing ZP summaries, zigzag persistence image Chen et al. (2021) and zigzag filtration curves Chen et al. (2022), can only deliver topological information for a single a-priori user-selected resolution scale, which restricts their utility for the scenarios with the yet unseen data. Here we bypass this major limitation and develop a new computationally efficient time-aware topological summary, *zigzag spaghetti*, simultaneously quantifying the most essential time-conditioned information for a sequence of graphs at all scales and opening a path for ZP in generative DL.

## 3 ZIGZAG SPAGHETTI: FROM TIME-AWARE KNOWLEDGE REPRESENTATION TO TOPOLOGICAL UNCERTAINTY QUANTIFICATION

Let $\mathcal{G} = (\mathcal{V}, \mathcal{E}, \boldsymbol{X})$ be an attributed graph, where $\mathcal{V}$ is a set of nodes ($|\mathcal{V}| = N$), $\mathcal{E}$ is a set of edges, and $\boldsymbol{X} \in \mathbb{R}^{N \times F}$ is a feature matrix of nodes (here $F$ is the dimension of the node features). Let $\boldsymbol{A} \in \mathbb{R}^{N \times N}$ be a symmetric adjacency matrix whose entries are $a_{ij} = \nu_{ij}$ if nodes $i$ and $j$ are connected and 0 otherwise (here $\nu_{ij}$ is an edge weight and $\nu_{ij} \equiv 1$ for unweighted graphs). In turn, $\boldsymbol{D}$ denotes the degree matrix of $\boldsymbol{A}$, that is $d_{ii} = \sum_j a_{ij}$. For spatio-temporal graph forecasting, a spatio-temporal graph is a collection of snapshots at different time steps, denoted by $\boldsymbol{\mathcal{G}} = \{\mathcal{G}^1, \mathcal{G}^2, \cdots, \mathcal{G}^{\mathcal{T}}\}$, where $\mathcal{T}$ is the maximum timestamp.

**Zigzag Persistent Homology** Given a sequence of time-evolving graphs $\mathcal{G}^{t_1}, \mathcal{G}^{t_2}, \ldots, \mathcal{G}^{t_n}$, we may be interested in such questions as:

- **Q1** *What are the most characteristic topological signatures of these graph sequence over time?*

- **Q2** *How do these time-aware topological signatures vary over different resolution scales?*

These questions are critical for forecasting, transfer learning, anomaly detection, and a broad range of other unsupervised tasks involving time-evolving objects.

To address these questions, we can invoke the machinery of zigzag persistence (ZP) (Carlsson & Silva, 2010; Carlsson et al., 2019), which allows us to consider linear maps into both directions $\mathscr{K}(\mathcal{G}_{\alpha_k}) \hookrightarrow \mathscr{K}(\mathcal{G}_{\alpha_{k+1}})$ and $\mathscr{K}(\mathcal{G}_{\alpha_k}) \hookleftarrow \mathscr{K}(\mathcal{G}_{\alpha_{k+1}})$, rather than into a single direction as traditional PH does (see Appendix A for background on PH). In a context of time-evolving graphs, we alternate left and right inclusions and interleave them with unions of neighboring graphs, where the union of graphs is defined as the standard graph operation by creating a single graph which contains all the nodes and edges from both original graphs (Gross et al., 2018; Shao & Sun, 2014):

$$\mathcal{G}^{t_1} \qquad \mathcal{G}^{t_2} \qquad \mathcal{G}^{t_3} \quad \ldots \quad \mathcal{G}^{t_{n-1}} \qquad \mathcal{G}^{t_n}$$

$$\searrow \quad \swarrow \quad \searrow \quad \swarrow \quad \searrow \quad \swarrow \quad \searrow \quad \swarrow$$

$$\mathcal{G}^{t_1} \cup \mathcal{G}^{t_2} \qquad \mathcal{G}^{t_2} \cup \mathcal{G}^{t_3} \qquad \ldots \qquad \mathcal{G}^{t_{n-1}} \cup \mathcal{G}^{t_n}$$

Now, given a scale $\alpha$, we say that a topological feature $\xi$ of dimension $p$ ($0 \leq p \leq dim(\mathcal{K})$) is born at time $t_b$, if it is first recorded at $\mathscr{K}(\mathcal{G}^{t_b})$, and we say that $\xi$ is born at time $t_b + \frac{1}{2}$, if it is first recorded at $\mathscr{K}(\mathcal{G}^{t_b} \cup \mathcal{G}^{t_b+1})$. Similarly, we say that $\xi$ dies at $t_d$ or $t_d + \frac{1}{2}$, if it is last recorded at $\mathscr{K}(\mathcal{G}^{t_d})$ or $\mathscr{K}(\mathcal{G}^{t_d} \cup \mathcal{G}^{t_d+1})$, respectively. The extracted topological features at scale $\alpha_k$ can be represented in a form of a zigzag persistent diagram (ZPD) $\text{PDz}_{\alpha_k}$, $k = 1, 2, \ldots, m$, which is a multiset in $\mathbb{R}^2$, i.e. $\text{PDz}_{\alpha_k} = \{(b_\xi, d_\xi) \in \mathbb{R}^2 | b_\xi < d_\xi, \xi \in \mathcal{M}\}$, where $b_\xi$ and $d_\xi$ are the birth and death of the $p$-dimensional topological feature $\xi$, respectively, $\mathcal{M}$ is a set containing the observed $p$-dimensional topological features at scale $\alpha_k$, and $m$ is a filtration length.

To quantify topological features extracted over time, Chen et al. (2021) and Chen et al. (2022) propose to use zigzag persistence images (ZPI) and zigzag filtration curves (ZFC), respectively. Both ZPI and ZFC are based on advancing the ideas of persistence images (Adams et al., 2017) and filtration curves (Johnson & Jung, 2021; O'Bray et al., 2021), developed for a traditional PH to a zigzag case. While promising, the major limitation of ZPI and ZFC is that these summaries are limited **only** to a single pre-defined resolution scale $\alpha_*$. In turn, selecting such feasible scale $\alpha_*$ may require some a-priori knowledge of the data. Furthermore, using ZPI and ZFC can help us answer **only a part** of Q1, that is, *which topological features are the most characteristic over time for a given resolution* $\alpha_*$? To mitigate this restriction, Xian et al. (2022) propose to employ a crocker plot, which does not use the notion of ZP, but uses the traditional PH framework, recording the number of "holes" at each scale $\alpha_i$ as a function of time $t$ and scales $\alpha_1, \alpha_2, \ldots$. While simple to compute, crocker plots do not satisfy the assumption of differentiability and, hence, cannot serve as an input to a fully trainable topological layer. In addition, crocker plots convey only local information on the graph topology, bypassing critical information on lifespans of topological features. Our goal is to address this major challenge and provide comprehensive answers both to Question 1 and 2 for a general case.

## 3.1 New Time-Aware Zigzag Spaghetti

Inspired by the ZFC of Chen et al. (2022) and crocker plots of Xian et al. (2022), we propose a new time-aware topological summary *zigzag spaghetti* (ZS). The term *spaghetti* is motivated by the notion of a spaghetti diagram widely used in Earth sciences (Wilks, 2011). ZS leverages the strengths of both ZFC and crockers plots, while mitigating their key limitations.

**Definition 3.1** (Zigzag Spaghetti). Let $[t_1, t_n]$ be the time interval over which we observe time evolving graphs $\{\mathcal{G}^t\}_{t_1}^{t_n}$. We represent $[t_1, t_n]$ as $\cup \Delta t_i$, where $\Delta t_i = (t_{i-1} + \frac{1}{2}, t_i), i = 1, \ldots, n$ are non-overlapping time intervals. Let $\alpha_1 < \alpha_2 < \ldots < \alpha_m$ be a sequence of scales. Then, Zigzag Spaghetti (ZS) for $p$-dimensional topological information of $\{\mathcal{G}^t\}_{t_1}^{t_n}$ ($0 \leq p \leq dim(\mathcal{K})$) is given by

$$
ZS\big(\{\mathcal{G}^t\}_t\big) = \begin{bmatrix} \sum_{j=1}^{\mathcal{M}} \omega_1 \kappa_1^{\alpha_1}(t_{b_j}, t_{d_j})_{\alpha_1} & \sum_{j=1}^{\mathcal{M}} \omega_2 \kappa_2^{\alpha_1}(t_{b_j}, t_{d_j})_{\alpha_1} & \cdots \sum_{j=1}^{\mathcal{M}} \omega_n \kappa_n^{\alpha_1}(t_{b_j}, t_{d_j})_{\alpha_1} \\ \vdots & \vdots & \vdots \\ \sum_{j=1}^{\mathcal{M}} \omega_1 \kappa_1^{\alpha_m}(t_{b_j}, t_{d_j})_{\alpha_m} & \sum_{j=1}^{\mathcal{M}} \omega_2 \kappa_2^{\alpha_m}(t_{b_j}, t_{d_j})_{\alpha_m} & \cdots \sum_{j=1}^{\mathcal{M}} \omega_n \kappa_n^{\alpha_m}(t_{b_j}.t_{d_j})_{\alpha_m} \end{bmatrix}
$$

Here $\kappa_i^{\alpha_k} : \mathbb{R}^2 \mapsto \mathbb{R}$ is a suitable Lipschitz continuous function with Lipschitz constant $L_i$, associated with scale $\alpha_k$, $k = 1, 2, \ldots, m$. Following Johnson & Jung (2021) and Chen et al. (2022), here as $\kappa_i$, we use a Gaussian density $f$ with mean $(t_{i-1} + 1/2, t_i)$ and identity covariance matrix $\Sigma$. In turn, $(t_{b_j}, t_{d_j})_{\alpha_k} \in \mathbb{R}^2$ is an interval containing the birth and death of a $p$-dimensional topological feature $\xi_j$ observed at scale $\alpha_k$, $j = \{1, 2, \ldots, \mathcal{M}\}$ and $\omega_i$ are positive weights such that $\sum_i \omega_i = 1$. (In our studies we set $\omega_i = 1/n$, i.e. a "flat prior".)

Being Lipschitz continuous, ZS is suitable as an input to a fully trainable topological layer in DL. At the same time, ZS inherits tractability of crocker plots in terms of linear algebra, while providing critical information on both local and global time-aware topology, lifespans of the extracted time-aware topological features at **all resolution scales**. Furthermore, as Proposition 3.2 shows, ZS satisfies the important theoretical stability guarantees in terms of Wasserstein distance $\mathcal{W}_1$.

**Proposition 3.2** (Stability of Zigzag Spaghetti). *Let* $\text{PDz}_{\alpha_k}$ *be a zigzag persistence diagram corresponding to scale* $\alpha_k$, $k = 1, 2, \ldots, m$, *and let* $PDz'_{\alpha_k}$ *be its perturbed counterpart such that* $\mathcal{W}_1\big(PDz_{\alpha_k}, PDz'_{\alpha_k}\big) < \epsilon_k$. *Let* $ZS$ *and* $ZS'$ *be zigzag spaghetti summaries corresponding to zigzag persistence diagrams* $PDz_{\alpha_k}$ *and* $PDz'_{\alpha_k}$, *respectively, over scales* $\alpha_1, \alpha_2, \ldots, \alpha_m$. *Then ZS is stable with respect to Wasserstein distance* $\mathcal{W}_1$ *and*

$$
||ZS - ZS'||_\infty \leq \max_{1 \leq k \leq m} \mathcal{W}_1\big(PDz_{\alpha_k}, PDz'_{\alpha_k}\big),
$$

*where $|| \cdot ||_\infty$ is a column norm of a matrix, i.e. for a matrix B, $||\mathbf{B}||_\infty = \max_i \sum_j |b_{ij}|$.*

Proof of Proposition 3.2 is in Appendix B.

Proposition 3.2 essentially guarantees that smaller changes in the graphs are expected to result in smaller changes in ZS. Practically, this stability result is of high importance to ensure robustness of ZS and the associated graph learning process with respect to uncertainties. To illustrate this idea, we conduct a robustness study under various noisy scenarios and varying sizes of training sets and find that ZS yields competitive resistance to a broad range of uncertainties (see Section 5 and Appendix D for more details). Furthermore, these ideas can be advanced toward a multipersistence framework, as discussed by Coskunuzer et al. (2024).

### 3.2 ZIGZAG TOPOLOGICAL UNCERTAINTY QUANTIFICATION

The idea of zigzag persistence is very general and can unfold new approaches to address a broad range of open problems in graph learning and knowledge representation. Here we propose to employ ZP for topological uncertainty quantification (UQ). That is, from a formal statistical inferential point, we can add another question:

- **Q3** *How certain are we that the extracted most characteristic topological signatures of the graph sequence over time and over different scales $\alpha_k$ are indeed most characteristic and are not simply due to a chance only?*

To address this question, we develop a bootstrap over ZS, which is rooted in the ideas of block bootstrap for time series Politis (2003). In particular, given a sequence of graphs $\{\mathcal{G}^t\}_{t_1}^{t_n}$ and its associated ZS, we (sub)sample $\tau_m$ graphs $\{\mathcal{G}^\tau\}_{\tau_1}^{\tau_n}$ without replacement out of $t_n$ graphs $\{\mathcal{G}^t\}_{t_1}^{t_n}$ ($\tau_m < t_n$) and construct its associated ZS*. We can then repeat the (sub)sampling procedure $B$ times, which results in an ensemble of bootstrapped ZS (BZS): BZS $= \{\text{ZS}_1^*, \text{ZS}_2^*, \dots, \text{ZS}_B^*.\}$ Intuitively, we can expect that the most illustrative time-aware topological features persisting over $\{\mathcal{G}^t\}_{t_1}^{t_n}$ shall also manifest in many bootstrapped ZS. Armed with the BZS ensemble, we can then consider integrating BZS into DL models, quantifying the uncertainty associated with time-aware topological signatures. Alternatively, we can estimate mean, median and other quantiles of BZS and construct the associated confidence and prediction intervals. However, this route is more challenging, since given that BZS is a collection of matrices, it involves the developments in terms of random matrix theory (Paul & Aue, 2014). As such, we leave this route for further research and note that the concept of zigzag and ZS are not restricted to time-evolving or even other naturally ordered objects.

## 4 ZIGZAG SPAGHETTI-AWARE NEURAL NETWORKS

In this section, we provide the technical details of the proposed zigzag spaghetti-aware models. Then detailed descriptions of each step are given.

### 4.1 MIXED-UP GRAPH CONSTRUCTION

To capture topological information from graph $\mathcal{G}$ and its node features, we construct a mixed-up graph $\mathcal{G}_\mathcal{M} = (\boldsymbol{A}_\mathcal{M}, \boldsymbol{X})$ based on original input graph $\mathcal{G}_\mathcal{O} = (\boldsymbol{A}_\mathcal{O}, \boldsymbol{X})$ and $k$-hop graph $\mathcal{G}_\mathcal{K} = (\boldsymbol{A}_\mathcal{K}, \boldsymbol{X})$.

**Original Graph Representation Learning.** We adopt the Graph Convolutional Layer (GCL) to perform message passing on the original graph $\mathcal{G}_\mathcal{O} = (\boldsymbol{A}_\mathcal{O}, \boldsymbol{X})$. It utilizes the original graph structure of $\mathcal{G}_\mathcal{O}$ with its node feature matrix $\boldsymbol{X}$ through the graph convolution operation and a multi-layer perceptron (MLP). Specifically, the designed graph convolution operation proceeds by multiplying the input of each layer with the $\tau$-th power of the normalized adjacency matrix. The $\tau$-th power operator contains statistics from the $\tau$-th step of a random walk on the graph (in this study, we set $\tau$ to be 2), thus nodes can indirectly receive more information from farther nodes in the graph. Combined with a multi-layer perceptron (MLP), the representation learned at the $\ell$-th layer is given by

$$\boldsymbol{\mathcal{Z}}_{\mathcal{G}_\mathcal{O}}^{(\ell+1)} = f_{\text{MLP}}(\sigma(\hat{\boldsymbol{A}}_\mathcal{O}^\top \boldsymbol{H}_{\mathcal{G}_\mathcal{O}}^{(\ell)} \boldsymbol{\Theta}^{(\ell)})). \tag{1}$$

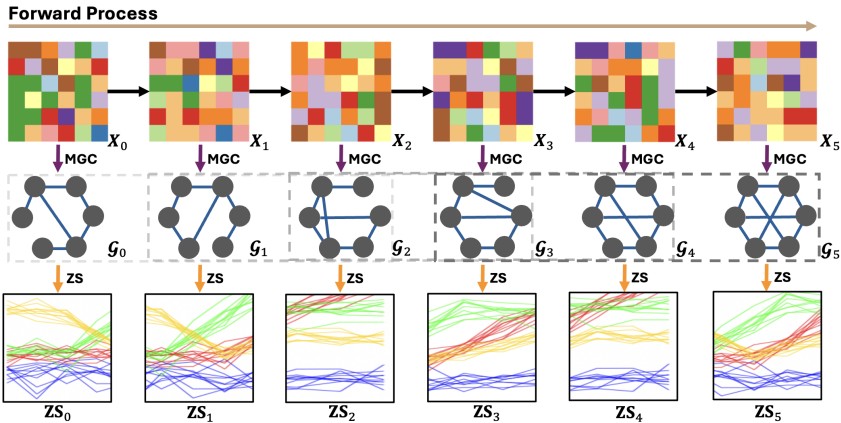

Figure 1: Forward process of the zigzag spaghetti-aware diffusion model. At each time step, we first add noise to the data, i.e., transitioning from $X_0$ to $X_1$ to $X_2$ and so on (in this toy example, we suppose there are overall 5 time steps). After that, we use the mixed-up graph construction (MGC) approach to generate the corresponding new mixed-up graph, and then apply the zigzag spaghetti method over the mixed-up graphs within a specific sliding window (i.e., the dashed gray box) to obtain the corresponding zigzag spaghetti.

Here $\hat{A}_{\mathcal{O}} = \tilde{D}_{\mathcal{O}}^{-\frac{1}{2}} \tilde{A}_{\mathcal{O}} \tilde{D}_{\mathcal{O}}^{\frac{1}{2}}$, $\tilde{A}_{\mathcal{O}} = A_{\mathcal{O}} + I$, $\tilde{D}$ is the corresponding degree matrix of $\tilde{A}$, $H_{\mathcal{G}_{\mathcal{O}}}^{(0)} = X$, $H_{\mathcal{G}_{\mathcal{O}}}^{(\ell)}$ is the output at the $(\ell-1)$-th layer, $f_{\text{MLP}}$ is an MLP which has 2 layers with batch normalization, $\sigma(\cdot)$ is the non-linear activation function, and $\Theta^{(\ell)}$ is a trainable weight of $\ell$-th layer.

**$\mathcal{K}$-Nearest Neighbor Graph Representation Learning.** First, to capture graph structural information, induced by the graph connectivity and node features, we build a $\mathcal{K}$-nearest neighbor ($\mathcal{K}$NN) graph, i.e., $\mathcal{G}_{\mathcal{K}} = (A_{\mathcal{K}}, X)$. In our study, we first define the similarity matrix $S_{\mathcal{K}} \in \mathbb{R}^{N \times N}$ among $N$ nodes and we consider three different methods as follows: *(i) Cosine Similarity*: It uses the cosine value of the angle be- tween two vectors to measure the similarity, i.e., $S_{uv} = \frac{x_u \cdot x_v}{|x_u||x_v|}$; *(ii) Gaussian Kernel*: It is based on the idea of the heat equation, a partial differential equation that describes how heat propagates over time $t$, which can be defined as follows $S_{uv} = \exp(-||x_u - x_v||^2/t)$; and *(iii) Node Embedding Similarity*: Let $H^{(\ell+1)}$ be the node embedding of $(\ell)$-th layer of GNN. For any $u, v \in \mathcal{V}$, we can calculate the similarity score $S_{uv}$ between nodes $u$ and $v$ as (i) Cosine Similarity: $S_{uv} = \frac{H_u^{(\ell+1)} \cdot H_v^{(\ell+1)}}{||H_u^{(\ell+1)}||||H_v^{(\ell+1)}||}$ or (ii) Gaussian Kernel: $S_{uv} = \exp(-||H_u^{(\ell+1)} - H_v^{(\ell+1)}||^2/t)$ (where $t$ is a free parameter). Then, the adjacency matrix $A_{\mathcal{K}}$ can be obtained through selecting top-$\mathcal{K}$ similar neighbouring nodes of each node. Similarly, we can use Eq. 1 to learn the $(\ell + 1)$-th layer node embeddings of the above $\mathcal{K}$NN graph, which is denoted by $\mathcal{Z}_{\mathcal{G}_{\mathcal{K}}}^{(\ell+1)}$.

**Mixup for Graph Construction.** Here we adopt node-level attention mechanism to learn the hidden connectivity between nodes. Specifically, given a node pair $(u, v)$, the importance coefficient between nodes $u$ and $v$ can be formulated as (for the simplicity, we omit $(\ell + 1)$ for $\mathcal{Z}_{\mathcal{G}_{\mathcal{O}}}^{(\ell+1)}$ and $\mathcal{Z}_{\mathcal{G}_{\mathcal{K}}}^{(\ell+1)}$):

$$e_{uv}^{\mathcal{M}} = \Theta_{\mathcal{M}}[\mathcal{Z}_{\mathcal{G}_{\mathcal{O}}}, \mathcal{Z}_{\mathcal{G}_{\mathcal{K}}}],$$
$$\alpha_{e_{uv}^{\mathcal{M}}} = \text{Softmax}(e_{uv}^{\mathcal{M}}) = \frac{\exp(\sigma(P_{\mathcal{M}} e_{uv}^{\mathcal{M}}))}{\sum_{v' \in \mathcal{V}} \exp(\sigma(P_{\mathcal{M}} e_{uv'}^{\mathcal{M}}))}, \quad (2)$$

where $[\cdot, \cdot]$ represents the concatenation operation, $\Theta_{\mathcal{M}}$ and $P_{\mathcal{M}}$ are training parameters, $\sigma(\cdot)$ denotes the LeakyReLU function with negative input slope as 0.1. After the above calculation, we can get the mixup attention score $\alpha_{e_{uv}^{\mathcal{M}}}$ which represents the weight of the edge between nodes $u$ and $v$.

### 4.2 ZIGZAG SPAGHETTI REPRESENTATION LEARNING

The challenge of graph representation learning in generative diffusion is further aggravated, when the goal is to model dynamic topological information between graphs. In light of this, we propose a novel

zigzag spaghetti-based encoder (ZS-ENC) to explicitly capture the beneficial temporal topological information. Formally, given ZS, we generate its latent representation as:

$$\boldsymbol{Z}_{\text{ZS}} = \begin{cases} f_{\text{MLP}}(\boldsymbol{\Theta}_{\text{ZS}}\text{ZS}), & \text{Scenario (I)} \\ f_{\text{MLP}}(\boldsymbol{\Theta}_{\text{ZS}}\text{BZS}), & \text{Scenario (II)} \end{cases}, \tag{3}$$

where $\boldsymbol{\Theta}_{\text{ZS}}$ is a trainable weight. Note that, in this paper, we consider both graph classification and prediction on spatio-temporal graphs, and we describe two specific scenarios to learn ZS, i.e.,

- **Scenario (I):** for spatio-temporal graph forecasting tasks, given a sequence of time-evolving graphs, we only generate one ZS;
- **Scenario (II):** for graph classification tasks, to generate a ZS for the sample $\boldsymbol{X}_t$ at time step $t$ in the forward process, we apply ZS over both time step $t$ and its adjacent $\varphi$ ($\varphi > 2$) time steps (e.g., $\{t - \varphi, \dots, t - 2, t - 1, t, t + 1, t + 2, \dots, t + \varphi\}$ - denoted as $\mathcal{T}_{t-\varphi:t+\varphi}$).

To get efficient computation without sacrificing on performance, we use the bootstrapping mechanism to randomly select $B$ subsamples from $\mathcal{T}_{t-\varphi:t+\varphi}$. For instance, we can randomly select $\rho(2\varphi + 1)$ (where $\rho \in [0.5, 1]$) time steps from $\mathcal{T}_{t-\varphi:t+\varphi}$ as $\{t-3, t-2, t-1, t, t+1\}$, $\{t-3, t-1, t, t+2, t+3\}$, etc. Hence, we can generate BZS with the size $B$ (in our study, we set $B$ to be $\{20, 50, 100\}$; see more details in Table 4) during the forward process of the diffusion model and each $\text{ZS}_b^*$ is generated based $\omega$ mixed-up graphs at $\omega$ different time steps.

### 4.3 ZIGZAG SPAGHETTI-AWARE DIFFUSION MODEL

We now elaborate on our two proposed frameworks. First, we start with zigzag spaghetti-aware diffusion model (ZS-DM) which involves forward and reverse processes for graph learning. Then, for spatio-temporal forecasting tasks, we discuss ZS-DM which combines the spatial and temporal learning capabilities of GNN, recurrent neural networks, and ZS representation learning module.

The forward process of the diffusion model is to gradually add noise onto the real data. We first generate a sample $\boldsymbol{X}_t$ from the input node feature $\boldsymbol{X}$ via Eq. 4. We then employ $f_{\text{MGC}}(\cdot)$ which is the mixed-up graph construction (MGC) process to generate its corresponding mixed-up graph, i.e., $\mathcal{G}_{\mathcal{M}}^t = f_{\text{MGC}}(\boldsymbol{X}_t)$. After that, in Eq. 5, we use the function of the ZS representation learning $f_{\text{ZS}}(\cdot)$ to extract the corresponding ZS:

$$\boldsymbol{X}_t = \sqrt{\overline{\alpha}_t}\boldsymbol{X}_0 + \sqrt{1 - \overline{\alpha}_t}\epsilon', \tag{4}$$
$$\boldsymbol{Z}_{\text{ZS},t} = f_{\text{ZS}}(f_{\text{MGC}}(\boldsymbol{X}_t)). \tag{5}$$

Inspired by incorporating directional noise in the forward diffusion process for graph learning of Yang et al. (2024), here we corrupt input data (i.e., $\boldsymbol{X}$) with directional noise instead of white noise, i.e., $\epsilon' = \text{sgn}(\boldsymbol{X}_0) \odot |\overline{\epsilon}|$ and $\overline{\epsilon} = \mu + \sigma \odot \epsilon$ where $\epsilon \sim \mathcal{N}(0, \boldsymbol{I})$, where $\odot$ is the elementwise product. During the mini-batch training, $\mu$ and $\sigma$ are calculated using graphs within the batch. The parameter $\overline{\alpha} = \prod_{s=0}^t (1 - \beta_s)$ represents the variance schedule ($\alpha_s = 1 - \beta_s$), and we set $\{\beta_1, \beta_2, \dots, \beta_T\}$ as hyperparameters so that the forward diffusion process is not included in the training. The overview of the forward process is illustrated in Figure 1.

In the denoising process, we develop a denoising decoder $f_{\text{DEC}}(\cdot)$ with graph convolution blocks (in a UNet-inspired architecture) to learn the reverse Markov chain with zigzag spaghetti:

$$\tilde{\boldsymbol{X}}_0 = f_{\text{DEC}}(\boldsymbol{X}_t, \boldsymbol{Z}_{\text{ZS},t}, t) = [f_{\text{GNN}}(\boldsymbol{X}_t), f_{\text{ZS}}(\boldsymbol{Z}_{\text{ZS-ENC},t})] + f_{\text{PE}}(t),$$

where we use sinusoidal position embeddings $f_{\text{PE}}(\cdot)$ (Vaswani et al., 2017) to encode the timestep $t$. For spatio-temporal graph prediction tasks, we employ UGnet Wen et al. (2023), i.e., an Unet-based architecture to capture temporal dependencies and the GNN to model spatial correlations.

## 5 EXPERIMENTS

**Datasets, Baselines, and Experimental Setup.** We evaluate our ZS-based graph learning models on spatio-temporal prediction tasks on 2 traffic datasets, i.e., PeMSD3 and PeMSD8 which are real-time traffic datasets from California (Guo et al., 2019; Song et al., 2020a), where nodes denote sensors and edges represent the intersections between the nodes.

Table 1: Prediction performance of probabilistic methods on the PeMSD3 and PeMSD8 datasets.

| Probabilistic Models | PeMSD3 | | | PeMSD8 | | |
|---|---|---|---|---|---|---|
| | MAE | RMSE | MAPE (%) | MAE | RMSE | MAPE (%) |
| Latent ODE Rubanova et al. (2019) | 17.25 | 28.33 | 17.71 | 26.05 | 39.50 | 17.20 |
| DeepAR Salinas et al. (2020) | 17.44 | 28.51 | 18.02 | 21.56 | 33.37 | 14.15 |
| CSDI Tashiro et al. (2021) | 18.92 | 30.41 | 19.56 | 32.11 | 47.40 | 18.88 |
| TimeGrad Rasul et al. (2021) | 17.93 | 29.81 | 19.33 | 24.46 | 38.06 | 17.03 |
| MC Dropout Wu et al. (2021) | 17.25 | 27.85 | 17.79 | 19.01 | 29.35 | 13.10 |
| DiffSTG Wen et al. (2023) | 17.79 | 28.74 | 18.12 | 18.60 | 28.20 | 11.94 |
| **ZS-DM (ours)** | **16.57** | **26.46** | **16.25** | **17.59** | **26.09** | **10.29** |

Table 2: Performance comparison on molecular and chemical graphs.

| Model | NCI1 | PROTEINS | DD | MUTAG | BZR | COX2 | PTC_MR | PTC_FM |
|---|---|---|---|---|---|---|---|---|
| GL Shervashidze et al. (2009) | N/A | N/A | N/A | 81.66±2.11 | N/A | N/A | 57.30±1.40 | N/A |
| WL Shervashidze et al. (2011) | 80.01±0.50 | 72.92±0.56 | 74.00±2.20 | 80.72±3.00 | N/A | N/A | 58.00±0.50 | N/A |
| DGK Yanardag & Vishwanathan (2015) | 80.31±0.46 | 73.30±0.82 | N/A | 87.44±2.72 | N/A | N/A | 60.10±2.60 | N/A |
| node2vec Grover & Leskovec (2016) | 54.89±1.61 | 57.49±3.57 | N/A | 72.63±10.20 | N/A | N/A | N/A | N/A |
| sub2vec Adhikari et al. (2018) | 52.84±1.47 | 53.03±5.55 | N/A | 61.05±15.80 | N/A | N/A | N/A | N/A |
| graph2vec Narayanan et al. (2017) | 73.22±1.81 | 73.30±2.05 | N/A | 83.15±9.25 | N/A | N/A | N/A | N/A |
| InfoGraph Sun et al. (2019) | 76.20±1.06 | 74.44±0.31 | 72.85±1.78 | 89.01±1.13 | 84.84±0.86 | 80.55±0.51 | 61.70±1.40 | 61.55±0.92 |
| GraphCL You et al. (2020) | 77.87±0.41 | 74.39±0.45 | 78.62±0.40 | 86.80±1.34 | 84.20±0.86 | 81.10±0.82 | 61.30±2.10 | 65.26±0.59 |
| AD-GCL Suresh et al. (2021) | 73.91±0.77 | 73.28±0.46 | 75.79±0.87 | 88.74±1.85 | 85.97±0.63 | 78.68±0.56 | 63.20±2.40 | 64.99±0.77 |
| RGCL Li et al. (2022) | 78.14±1.08 | 75.03±0.43 | 78.86±0.48 | 87.66±1.01 | 84.54±1.67 | 79.31±0.68 | 61.43±2.50 | 64.29±0.32 |
| GCL-TAGS Lin et al. (2022) | 71.43±0.49 | 75.78±0.41 | 75.78±0.52 | 89.12±0.76 | N/A | N/A | N/A | N/A |
| GraphMAE Hou et al. (2022) | 80.40±0.30 | 75.30±0.39 | N/A | 88.19±1.26 | N/A | N/A | N/A | N/A |
| CWN Bodnar et al. (2021) | 80.16±0.35 | 72.51±0.74 | N/A | 86.32±0.91 | N/A | N/A | N/A | N/A |
| TOGL Horn et al. (2022) | 78.59±0.47 | 72.22±0.79 | N/A | 90.49±0.76 | N/A | N/A | N/A | N/A |
| DDM Yang et al. (2024) | 73.93±0.77 | 75.47±0.50 | N/A | 91.51±1.45 | 83.64±0.80 | 79.88±0.34 | 62.11±2.35 | 65.09±0.97 |
| **ZS-DM (Ours)** | **81.80±0.09**[*] | **76.08±0.19**[*] | **78.93±0.32** | **91.68±0.34** | **86.20±0.12**[*] | **81.73±0.86**[*] | **64.02±1.00** | **66.76±0.24**[*] |

In addition, we validate our ZS-DM on unsupervised graph representation learning classification tasks using the following 8 real-world chemical compounds and protein molecules: (i) 3 chemical compound datasets: MUTAG, BZR, and COX2, where the graphs are chemical compounds, the nodes are different atoms, and the edges are chemical bonds and (ii) 5 molecular compound datasets: NCI1, D&D, PROTEINS, PTC_MR, and PTC_FM, where the nodes represent amino acids and edges represent relationships or interactions between the amino acids, e.g., physical bonds, spatial proximity, or functional interactions. For these 8 graphs, we follow the training principle (You et al., 2020) and use 10-fold cross validation accuracy as the classification performance (based on a non-linear SVM model, i.e., LIB-SVM Chang & Lin (2011)) and report the mean and standard deviation. We conduct a one-sided two-sample $t$-test between the best result and the best performance achieved by the runner-up, where $*$ denotes statistically significant results. We also adopt a larger graph dataset ogbg-molhiv from Open Graph BenchMark (OGB) Hu et al. (2020). For ogbg-molhiv data, the task is to predict a certain molecular property, measured in terms of Receiver Operating Characteristic Area Under Curve (ROC-AUC) scores, and we follow the official scaffold splitting where structurally different molecules are separated into different subsets.

**Experimental Settings.** We implement our proposed ZS-DM with Pytorch framework on two NVIDIA RTX A5000 GPUs with 24 GB RAM. We use the dionysus2 package in Python for ZP on graphs. For graph classification, we follow a two-step process, i.e., we firstly pre-train a ZS-DM on the dataset in an unsupervised manner, and then extract feature representations from diffusion steps 50, 100, and 200 using the pre-trained model. For PeMSD3 and PeMSD8, we consider window size

Table 3: Ablation study of different zigzag-based topological features.

| Architecture | PROTEINS | MUTAG | PTC_MR |
|---|---|---|---|
| **ZS-DM** | **76.08±0.19** | **91.68±0.34**[*] | **64.02±1.00** |
| ZPI-DM | 75.83±0.41 | 89.28±0.90 | 63.06±1.32 |
| ZFC-DM | 75.80±0.49 | 90.74±0.31 | 62.43±1.43 |

Table 4: Performance comparison under different number of subsampling replications on PROTEINS, MUTAG, and PTC_MR datasets.

| Model | PROTEINS | | | MUTAG | | | PTC_MR | | |
|---|---|---|---|---|---|---|---|---|---|
| | # sim = 20 | # sim = 50 | # sim = 100 | # sim = 20 | # sim = 50 | # sim = 100 | # sim = 20 | # sim = 50 | # sim = 100 |
| ZS-DM (ours) | 75.98±0.54 | 76.02±0.28 | **76.08±0.19** | 91.62±0.57 | 91.48±0.50 | **91.68±0.34** | 63.98±2.28 | 63.33±1.07 | **64.02±1.00** |

$\omega$ of 12 and horizon $h$ of 12. To measure the forecasting performance of predictive models, we use Mean Absolute Error (MAE), Root Mean Square Error (RMSE), and Mean Absolute Percentage Error (MAPE). The hyperparameter values is determined via grid search. For spatio-temporal traffic forecasting, we set the number of epochs trained, batch size, and weight decay ratio to be 200, 16, and 0.9 respectively. We search initial learning rate of model among $\{0.001, 0.003, 0.005, 0.01\}$, number of convolution layers among $\{1, 2, 3\}$, dimension of hidden units among $\{16, 32, 64, 128, 512\}$. For the mixed-up graph construction, we set number of steps of a random work to be $\tau = 2$. Our code and data can be accessed under `https://github.com/zigzagspaghetti/zigzag_spaghetti_dm.git`. (Please refer to Appendix C for the detailed version of baselines.)

**Findings.** Table 1 compares forecasting performance on spatio-temporal graphs between our ZS-DM and state-of-the-art baselines (6 probabilistic methods) on PeMSD3 and PeMSD8. We find that ZS-DM leads performance under all scenarios, with gains up to 14% in MAPE and up to 7.5% in RMSE. The results suggest that ZS-DM can effectively capture spatial correlations, temporal dependencies, and time-aware topological information in a holistic manner.

In turn, Table 2 presents accuracy for graph classification. The best results are in bold and the results with underline denote the runner-ups. We find that ZS-DM consistently outperforms 15 baselines on all 8 graphs. To be specific, compared to GNN-based contrastive learning (i.e., InfoGraph, GraphCL, AD-DCL, RGCL, and GCL-TAGS), ZS-DM improves upon runner-ups by a margin of 4.68%, 0.40%, 0.09%, 2.87%, 0.27%, 0.90%, 1.30%, and 2.30% on NCI1, PROTEINS, DD, MUTAG, BZR, COX2, PTC_MR, and PTC_FM respectively. Moreover, compared to the two powerful graph generative models (i.e., GraphMAE and DDM), ZS-DM al-

Table 5: Performance comparison on ogbg-molhiv (ROC-AUC).

| Model | ogbg-molhiv |
|---|---|
| GraphCL | 65.18±2.53 |
| TOGL | 62.63±2.39 |
| **ZS-DM (ours)** | **67.52±3.00** |

ways outperforms the runner-up by a large margin. For instance, for chemical compound datasets (except MUTAG), ZS-DM demonstrates competitive improvement, i.e., yielding a relative gain from 2.32% to 3.06%. The results of ZS-DM, GraphCL, and TOGL on ogbg-molhiv are shown in Table 5. The findings are consistent with those in Table 2, that is, ZS-DM yields better performance than that of GraphCL and TOGL.

**Robustness.** We also conduct a robustness of ZS-DM under various noisy conditions. In particular, we add Gaussian noise with mean $\mu = 0$ and standard deviation $\sigma = 0.1$ to 1% and 5% MUTAG data. As Table 6 indicates, ZS-DM is more stable under noisy scenarios than DDM.

We perform the robustness analysis w.r.t. varying sizes of the training sets, reducing the training set from 90% to 80% and 70% (see Appendix D for further details). We find that the ZS-DM gains in performance increase as the training size decreases, while variability of ZS-DM tends to be noticeably lower than runner up SOTAs, i.e., up to 1.5-2 times lower. These phenomena suggest that ZS might be more helpful when the amount of training data is lower or the data are noisy, which intuitively is expected because ZS allows us to gain additional insights into the latent topological structure of the underlying data generating process. (For additional experiments on the choice of filtration scales and sensitivity to the dimensions of topological features see Appendix D also contains.)

Table 6: Robustness study under additive Gaussian noise.

| Model | MUTAG | MUTAG with 1% noise | MUTAG with 5% noise |
|---|---|---|---|
| DDM | 91.51±1.45 | 89.68±0.70 | 86.41±0.76 |
| **ZS-DM (ours)** | **91.68±0.34** | **91.02±0.83** | **89.76±0.36** |

**ZS vs. competing Zigzag Summaries and Traditional Persistence.** To evaluate the ZS gains in capturing time-aware higher-order topological information, we compare ZS with ZPI Chen et al.

(2021) and ZFC Chen et al. (2022). Table 3 indicates that in all cases (both graph prediction and graph classification), ZS-DM outperforms ZPI-/ZFC-based models by a large margin. Also, compared with ZPI, computational cost of ZS is much lower, e.g., average running time of ZS and ZPI generation per epoch on MUTAG are 0.21 and 0.37 seconds, respectively. To summarize, our proposed approach achieves much better results than ZPI- and ZFC-based DM models without sacrificing efficiency.

Furthermore, Table 7 presents comparison of ZS vs. traditional PH. We observe that ZS-DM outperforms with highly statistically significant gains the diffusion model with traditional persistence on both MUTAG and BZR datasets. These phenomena illustrate the critical role that simultaneous assessment of the *joint* higher order topological features at all resolution scales plays for performance and robustness of graph diffusion models.

Table 7: Performance comparison between ZS and traditional persistence.

| Data | ZS | Traditional Persistence |
|------|------|-------------------------|
| MUTAG | **91.68±0.34*** | 86.00±0.83 |
| BZR | **86.20±0.12*** | 83.94±0.37 |

**Impact of the Number of Subsampling Replications and Topological UQ.** As described in Section 4.2, in the forward process of ZS-DM model, we can generate a BZS of size $B$. To evaluate the ZS-DM performance with different bootstrap samples, we also report the mean accuracy with standard deviation of ZS-DM for for varying numbers of bootstrap replications (i.e., #sim) (see Table 4). We observe that, when #sim increases (from 20 to 100), variability of ZS-DM substantially decreases, thereby suggesting the utility of this approach for topological UQ.

**Computational Costs.** Currently, the best possible computational complexity for ZP for 0-dimensional and for 1-dimensional features on graphs is $O(m\log^2(N) + m\log(m))$ Dey & Hou (2021). The ZP complexity on graphs can be improved for a subclass of graphs satisfying certain assumptions on the size of the unions in the filtration, i.e. of the size of the union of all graphs in the filtration is $\Omega(m^\epsilon)$ for any fixed $0 < \epsilon < 1$ Dey et al. (2023). Alternatively, we may compute ZP based only on landmarks rather than on all nodes and then to use Dowker or witness complexes (De Silva & Carlsson, 2004; Choi et al., 2024; Li et al., 2024). Table 8 also shows average time taken and performance comparison between ZS-DM and DDM. Although the average time taken by ZS-DM is a bit higher, ZS-DM consistently outperforms DDM and other competitors.

Table 8: Average time taken comparison between ZS-DM and baseline methods.

| Data | MUTAG | PTC_MR | | Data | PeMSD3 | PeMSD8 |
|------|-------|--------|---|------|--------|--------|
| DDM | 1.73 s | 3.41 s | | DiffSTG | 3.98 s | 4.73 s |
| **ZS-DM (ours)** | 2.55 s | 3.79 s | | **ZS-DM (ours)** | 3.27 s | 5.25 s |

# 6 DISCUSSION AND FUTURE DIRECTIONS

With the growing success of diffusion models on graphs, there is a surge of interest in more accurate and reliable graph representation learning. Indeed, most currently prevailing techniques of generative diffusion on graphs tend to be limited in their abilities to capture intrinsic topological features of *multiple graphs simultaneously*. This in turn severely obstructs the transferability and generalizability of such diffusion models. We tackle this fundamental challenge by invoking the mathematical machinery of zigzag persistence which allows us to systematically extract and summarize the inherent topological characteristics of multiple objects, including time-evolving graphs, simultaneously at multiple resolution scales in a form of *zigzag spaghetti*. To the best of our knowledge, this is the first attempt to bridge diffusion models on graphs with the notions of topological data analysis and, zigzag persistent homology, in particular. In the future, armed with more scalable abstract simplicial complexes such as Dowker and witness complexes, we plan to push this envelop further by exploring zigzag persistence for diffusion models on hypergraphs and dynamic multilayer networks.

## ACKNOWLEDGMENTS

This work was supported by the NSF grant DMS-2335846/2335847 and the ONR grant N00014-21-1-2530. The views expressed in the article do not necessarily represent the views of NSF or ONR. The authors would like to thank the AC and three ICLR reviewers as well Dr. Andrei Zagvozdkin for engaging into the interactive discussion and providing highly valuable feedback that allowed for improving the manuscript.

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

## A  BACKGROUND ON TRADITIONAL PERSISTENCE

Persistent homology (PH) is a mathematical machinery which allows us to quantify shape of an object along various dimension(s). By shape here, we broadly understand object properties that are preserved under continuous transformations, i.e. the ones that do not change "holes" in the object. Such transformations include, for example, folding, bending, and twisting; while "holes" can be connected components (0-dim topological features), loops (1-dim topological features), voids (2-dim topological features), and their higher-dimensional counterparts. PH monitors evolution of these topological features as we monotonically change certain user-defined parameters. The idea is to select some suitable monotonic sequence of scales $\alpha_1 < \alpha_2 < \ldots < \alpha_m$, and then to study graph $\mathcal{G}$ not as a single object but as a parametrized sequence of nested graphs $\mathcal{G}_{\alpha_1} \subseteq \mathcal{G}_{\alpha_2} \subseteq \ldots \mathcal{G}_{\alpha_m} = \mathcal{G}$, recording which holes appear (born) and disappear (die) throughout this filtration. Holes that persist over the filtration (i.e., with longer lifespans) are likelier to contain some essential latent information about the structural organization of $\mathcal{G}$, while holes with shorter lifespans are often referred to as topological noise. To systematize the process of selecting and recording holes, we build an abstract simplicial complex $\mathcal{K}(\mathcal{G}_{\alpha_i})$ on each $\mathcal{G}\alpha_i$, which results in a filtration of complexes $\mathcal{K}(\mathcal{G}_{\alpha_1}) \subseteq \mathcal{K}(\mathcal{G}_{\alpha_m}) \subseteq \ldots \mathcal{K}(\mathcal{G}_{\alpha_m})$ and counting the associated simplices in $\mathcal{K}$. As an example of such scale parameter $\alpha$, we can choose edge weight and then use an abstract simplicial complex $\mathcal{K}(\mathcal{G}_\alpha) = \{\tilde{\mathcal{G}} \subseteq \mathcal{G} | diam(\tilde{\mathcal{G}}) \leq \alpha\}$, that is, we keep only nodes (and the associated induced subgraphs) with a shortest weighted path of at most $\alpha$. Such $\mathcal{K}$ is called a Vietoris-Rips complex and the induced nested sequence is called a power filtration. (For an overview of various filtration on graphs see Adams et al. (2017); Bauer (2019); Hofer et al. (2020).) As such, this traditional PH framework considers a sequence of linear maps $\mathcal{K}(\mathcal{G}_{\alpha_i}) \hookrightarrow \mathcal{K}(\mathcal{G}_{\alpha_{i+1}})$ along the same direction, thereby assuming that each simplicial complex needs to be a subset of the following one, allowing us **only** to add new simplices to the preceding complex. This limits applicability of PH to time-dependent and dynamic graphs, where the goal is to extract time-aware topological signatures that persist over time.

## B  PROOF OF PROPOSITION 5.2

*Proof.* By definition of ZS, we get

$$||ZS - ZS'||_\infty = \max_{1 \leq k \leq m} \sum_{l=1}^{n} \left| ZFC_{\alpha_k}(\Delta t_l) - ZFC'_{\alpha_k}(\Delta t_l) \right|,$$

where

$$ZFC_{\alpha_k}(\Delta t_l) = \sum_{j=1}^{\mathcal{M}} \omega_l \kappa_l^{\alpha_k}(t_{b_j}, t_{d_j})_{\alpha_k},$$

corresponding to zigzag persistence diagram $\text{PDz}_{\alpha_k}$ and $ZFC'_{\alpha_k}(\Delta t_l)|$ is its counterpart corresponding to the perturbed $\text{PDz}'_{\alpha_k}$.

From Proposition 3.2 of Chen et al. (2022), we have

$$\left| ZFC_{\alpha_k}(\Delta t_l) - ZFC'_{\alpha_k}(\Delta t_l) \right| \le w_l L_l \mathcal{W}_1\big(\text{PDz}_{\alpha_k}, \text{PDz}_{\alpha_k}\big) < w_l L_l \epsilon_k.$$

Hence,

$$||ZS - ZS'||_\infty \le \max_{1 \le k \le m} \sum_{l=1}^{n} w_l L_l \mathcal{W}_1\big(\text{PDz}_{\alpha_k}, \text{PDz}_{\alpha_k}\big)$$

$$\le L \max_{1 \le k \le m} \mathcal{W}_1\big(\text{PDz}_{\alpha_k}, \text{PDz}_{\alpha_k}\big) \left( \sum_{l=1}^{n} \omega_l \right)$$

$$\le L \max_{1 \le k \le m} \mathcal{W}_1\big(\text{PDz}_{\alpha_k}, \text{PDz}_{\alpha_k}\big) \le L\epsilon,$$

where $L = \max 1 \le l \le n L_l$ and $\epsilon = \max_{1 \le k \le m} \epsilon_k$. $\qquad\square$

## C    BASELINES AND EXPERIMENTAL SETUP

**Baselines.** We use the following popular models for spatio-temporal graph forecasting as baselines: (i) 6 probabilistic methods: (1) Latent Ordinary Differential Equations (ODE) Rubanova et al. (2019), (2) DeepAR (which is a forecasting method based on autoregressive recurrent neural networks) Salinas et al. (2020), (3) Conditional Score-based Diffusion models for Imputation (CSDI) Tashiro et al. (2021), (4) TimeGrad (which is an autoregressive model for multivariate probabilistic time series forecasting) Rasul et al. (2021), (5) MC Dropout (which uses the MC Dropout Gal et al. (2017) for probabilistic spatio-temporal forecasting) Gal et al. (2017); Wu et al. (2021), and (6) DiffSTG (which is a non-autoregressive framework) Wen et al. (2023); and (ii) 6 GNN-based models: (1) Spatio-Temporal Graph Convolutional Networks (STGCN) Yu et al. (2018), Diffusion Convolutional Recurrent Neural Network (DCRNN) Li et al. (2018), Adaptive Graph Convolutional Recurrent Network (AGCRN) Bai et al. (2020), Spectral Temporal Graph Neural Network (StemGNN) Cao et al. (2020), STID Shao et al. (2022), and Fourier Graph Neural Network (FourierGNN) Yi et al. (2024b). For graph classification, we compare our ZS-DM with 15 state-of-the-art (SOTA) baselines including: (1) Graphlet Kernel (GL) Shervashidze et al. (2009), (2) Weisfeiler-Lehman Sub-tree Kernel (WL) Shervashidze et al. (2011), (3) Deep Graph Kernels (DGK) Yanardag & Vishwanathan (2015), (4) node2vec Grover & Leskovec (2016), (5) sub2vec Adhikari et al. (2018), (6) graph2vec Narayanan et al. (2017), (7) InfoGraph Sun et al. (2019), (8) Graph Contrastive Learning (GraphCL) You et al. (2020), (9) Adversarial-Graph Contrastive Learning (AD-GCL) Suresh et al. (2021), (10) Rationale-aware Graph Contrastive Learning (RGCL) Li et al. (2022), (11) Graph Contrastive Learning scheme with Topology Augmentation guided by the Graph Spectrum (GCL-TAGS) Lin et al. (2022), (12) Masked Graph Autoencoder (GraphMAE) Hou et al. (2022), (13) CW Networks (CWN) Bodnar et al. (2021), (14) Topological Graph Neural Networks (TOGL) Horn et al. (2022), and (15) Directional Diffusion Models (DDM) Yang et al. (2024).

## D    ROBUSTNESS AND SENSITIVITY ANALYSIS

**Robustness to Noise and Varying Sizes of Training Data** We also perform experiments on reducing the training set from 90% to 80% and 70%. We find that the ZS-DM gains are higher for lower training sizes, i.e. on average for MUTAG and BZR 0.22% for 90% training size (Table 2 main body), 1.58% for 80% training size (Table 9 below), and 1.94% for 70% training size (Table 10 below). Furthermore, variability of ZS-DM tends to be noticeably lower than runner up SOTAs, i.e., up to 1.5-2 times lower, suggesting that the ZP idea may be helpful for uncertainty quantification. More

Table 9: Performance comparison of robustness study with 80% training set.

| Model | MUTAG | BZR |
|---|---|---|
| AD-GCL | 82.16±2.09 | 80.43±1.55 |
| RGCL | 82.77±2.34 | 81.35±2.00 |
| DDM | 83.38±2.85 | 81.21±1.85 |
| **ZS-DM (ours)** | **84.59±1.47** | **82.78±1.29** |

Table 10: Performance comparison of robustness study with 70% training set.

| Model | MUTAG | BZR |
|---|---|---|
| AD-GCL | 79.50±2.23 | 74.28±2.22 |
| RGCL | 80.35±1.68 | 75.45±1.85 |
| DDM | 80.78±2.07 | 76.92±2.88 |
| **ZS-DM (ours)** | **82.23±1.57** | **78.58±1.04** |

generally, these findings suggest that the ZS approach is capable of capturing some fine-grained latent information on the graph structure that the more conventional approaches cannot, but the role of such finer-grained information diminishes as the sample size increases.

Table 11: Sensitivity of graph classification with different choices of scales.

| Data | degree | betweenness | closeness |
|---|---|---|---|
| MUTAG | **91.68±0.34**[*] | 90.70±0.83 | 88.40±0.37 |
| BZR | **86.20±0.12**[*] | 83.35±1.93 | 82.52±0.82 |

Table 12: Performance comparison among ZS features with different dimensions.

| Data | 0-dim | 1-dim | 0- & 1-dim |
|---|---|---|---|
| MUTAG | **91.68±0.34**[*] | 88.91±0.47 | 90.17±0.12 |
| BZR | **86.20±0.12**[*] | 84.34±0.90 | 85.30±0.36 |

**Sensitivity Analysis to the Choice of Filtration Scales.** Table 11 shows the sensitivity of graph classification results to the choice of scales, i.e., using node-degree, node-betweenness, and node-closeness scores. In general, we find that the performance for more homogenous graphs is less sensitive to filtrations. For sparser and more heterogeneous graphs, degree-based or power filtrations are often the preferred choices. We have also conducted additional experiments to compare ZS and traditional persistence. From Table 7, we observe that our diffusion model with ZS always outperforms with highly statistically significant gains the diffusion model with traditional persistence on both MUTAG and BZR datasets. Additionally, in experiments of graph classification (see Table 12), we can incorporate 0-dimensional feature, 1-dimension feature, or both 0- and 1-dimensional features into the model. However, we found that our ZS-DM model with 0-dimensional features always outperforms other scenarios. This phenomenon can be potentially explained by stronger signal yielded by 0-dimensional features and much higher representation of 0-dimensional features.

## E  ADDITIONAL DETAILS OF ZIGZAG SPAGHETTI AND EXPERIMENTS

Zigzag spaghetti has two main advantages: first, it allows for capturing essential topological characteristics of a dynamic object over time at **all resolution scales simultaneously**, and, second, it is easily tractable and computationally efficient (relative to other time-aware topological summaries). To justify the second point, we present the running time comparison among ZS, ZPI, and ZFC for graph classification on MUTAG. The obtained results suggest that ZS-DM achieves the best performance and also delivers competitive running time (in seconds):

Table 13: Running time and performance comparison.

| Data | ZS-DM (ours) | ZPI-DM | ZFC-DM |
|---|---|---|---|
| MUTAG | 0.21 sec (91.68±0.34) | 0.37 sec (90.52±0.63) | 0.18 sec (90.73±0.59) |

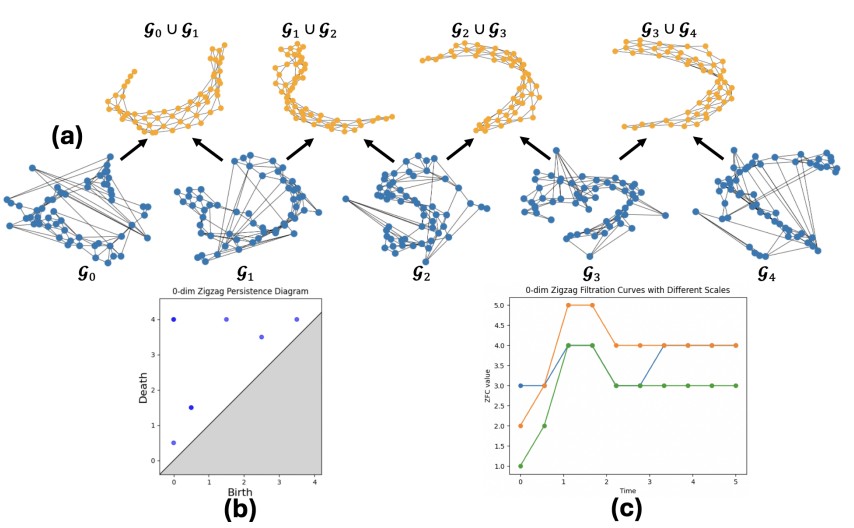

Figure 2: Visualization of zigzag persistence, zigzag persistence diagram and zigzag filtration curves.

In addition, the complexity of zigzag persistence on graphs can be improved for a subclass of graphs satisfying certain assumptions on the size of the unions in the filtration, i.e. of the size of the union of all graphs in the filtration is $\Omega m^\epsilon$ for any fixed $\epsilon$, i.e., $0 < \epsilon < 1$ Dey & Hou (2023). However, this is not true for a general class of graphs, and some other more drastic approaches are needed to improve scalability of zigzag persistence on denser graphs. One such potential direction is to compute zigzag persistence based only on landmarks rather than on all nodes and then to use Dowker or witness complexes. This approach has not been yet investigated in computational topology and its theoretical guarantees are yet unknown. We believe though that it is a very promising approach that can fundamentally shift the scalability problem for ZP.

Furthermore, based on Qin et al. (2023), we incorporate its sparse denoising network (which contains a graph transformer architecture with a sparse attention mechanism) into our zigzag spaghetti approach and have conducted additional experiments on MUTAG and BZR datasets. We call the zigzag spaghetti model with sparse denoising network ZS-DM$_{new}$. From the Table 14, we observe that ZS-DM$_{new}$ always outperforms the previous version (i.e., ZS-DM$_{old}$) on both MUTAG and BZR datasets. Thanks very much for this excellent suggestion for further improvement of the ZS approach.

From the Figure 2 (c), we observe that although overall there are similarities in curves for all scales, we find that, starting from time 3, the time evolving graphs deliver consistency in terms of topological characteristics of order 1, while there is a notable variability among topological characteristics over the time period from time point 0 to time point 3. Furthermore, another notable topological feature is observed over the time period 1-2, but its lifespan is relatively short.

Table 14: ZS-DM with sparse denoising network.

| Model | MUTAG | BZR |
|---|---|---|
| ZS-DM$_{old}$ | 91.68±0.34 | 86.20±0.12 |
| ZS-DM$_{new}$ | 92.15±0.32 | 86.39±0.10 |

