# OpenReview forum: "Topological Zigzag Spaghetti for Diffusion-based Generation and Prediction on Graphs"
_ICLR.cc/2025/Conference — ICLR 2025 Poster_

### Official Review · Reviewer_MZ8J · 2024-10-30

**Soundness:** 3
**Presentation:** 3
**Contribution:** 3
**Rating:** 6
**Confidence:** 3

**Summary:**

This paper addresses the a critical limitation in current graph diffusion models by introducing a novel approach to capture higher-order topological properties. Such higher-order topological properties can be captured using time-aware topological summary zigzag spaghetti (ZS), which leverages zigzag persistent homology. Stability of the obtained ZS has been guaranteed with respect to Wasserstein distance. The paper also introduces zigzag spaghetti-based encoder (ZS-ENC) to incorporate ZS into diffusion models. The applicability of ZS has been tested on various settings demonstrating improvements in different metrics.

**Strengths:**

- This paper attempts to incorporate zigzag persistence homology into generative diffusion model.
- Theoretical stability of zigzag spaghetti has been addressed
- Applicability of zigzag spaghetti has been demonstrated in different settings

**Weaknesses:**

The soundness of the paper seems to need a bit of improvement. For instance, notations might need a little more description. ($\omega$ from section 3 and 3.1 seems to be referring to different object). Also, definition of mathematical object can help the understanding of the paper (union of two graphs mentioned). More detailed description of figures and tables can improve the readability.

**Questions:**

- What is the difference between $\mathcal{G}_t$ and $\mathcal{G}^t$ in Sec. 3?
- Can you describe how you obtained $\omega_i$ in Sec. 3.1
- How is Sec. 4.1 related to the methodology of your paper?

---

> ### Author Response · Authors · 2024-11-19
> **Rebuttal by Authors**
>
> We appreciate very much your constructive comments on our paper. Please kindly find our response to your comments below, and all revisions made to the paper are highlighted in red.
>
> $\textbf{Q1:}$ The soundness of the paper seems to need a bit of improvement. For instance, notations might need a little more description. ($\omega$ from section 3 and 3.1 seems to be referring to different object). Also, definition of mathematical object can help the understanding of the paper (union of two graphs mentioned). More detailed description of figures and tables can improve the readability.
>
> $\textbf{A:}$ Thank you very much for careful reading! Yes, we used $\omega_{.,.}$ with 2 subscripts to denote an edge weight and  $\omega_{.}$ with 1 subscript to denote a weight in zigzag spaghetti. We agree that it may be confusing so we have changed the edge weight to $\nu$ in the paper.
>
> On the graph unions, we use the standard graph operation, where the union of graphs is defined by creating a single graph containing all the nodes and edges from both original graphs (Gross et al. (2018) “Graph Theory and Its Applications”, CRC Press; Shiu and  Sun (2014) “A First Course in Graph Theory”, HKBU). We have added the respective explanation and the references to the paper.
>
>
> $\textbf{Q2:}$ What is the difference between $G_t$ and $G^t$ in Sec. 3?
>
> $\textbf{A:}$ This is an inconsistency on our part, thanks for noticing it! $G_t$ (with subscript) and $G^t$ (with superscript) is the graph observed at point $t$. We have updated the notation in the paper to $G^t$ to denote dependence of graphs on time.
>
> $\textbf{Q3:}$ Can you describe how you obtained $w_i$ in Sec. 3.1?
>
> $\textbf{A:}$We use here a standard approach of assigning $w_i=1/n$, that is, all time intervals contribute equally (which would correspond to flat priors in Bayesian terms). However, this raises interesting directions of either learning optimal $w_i$ with attention mechanisms or using more refined priors with the Bayesian framework. Thanks very much for initiating these ideas!
>
> $\textbf{Q4:}$ How is Sec. 4.1 related to the methodology of your paper?
>
> $\textbf{A:}$ The Sec. 4.1 is about the mixed-up graph construction (MGC) which is used to generate a mixed-up graph $\mathcal{G}^t$ based on a sample ${\bf X}_t$ (see Eq. 5) where ${\bf X}_t$ corrupted node feature matrix in the forward process at time step $t$. Then we apply our zigzag spaghetti approach to a mixed-up graph $\mathcal{G}^t$ at time step $t$ and obtain a corresponding zigzag spaghetti-based topological feature which will be used to integrate into the denoising decoder.

---

> > ### Author Response · Authors · 2024-11-20
> > **Thanks so much!**
> >
> > Thank you very much for the careful reading of the paper, constructive feedback and, of course, for raising the score!

---

> > ### Comment · Reviewer_MZ8J · 2024-11-25
> >
> > I appreciate your thoughtful response and the effort to improve the paper's readability. The revisions effectively highlight the novelty of the work, making the contributions more apparent. Additionally, the inclusion of the reported results and the comprehensive responses to other reviewers' concerns were compelling and have positively influenced my evaluation of the paper's soundness, presentation, and overall contribution. Thank you for presenting such an innovative approach.

---

> > > ### Author Response · Authors · 2024-11-25
> > > **Thanks very much!**
> > >
> > > Thanks so much for all the motivating feedback and encouragement, Reviewer MZ8J!!!

---

### Official Review · Reviewer_HF5K · 2024-10-30

**Soundness:** 3
**Presentation:** 3
**Contribution:** 2
**Rating:** 5
**Confidence:** 3

**Summary:**

This paper investigates diffusion models on graph generation and prediction. To capture the high-order topological graph properties during diffusion, the authors propose to leverage the zigzag persistence. In particular, a computation-efficient topological summary metric zigzag spaghetti (ZS) is developed over a sequence of graphs at multiple resolutions with theoretical stability guarantees. Experimental results confirm the effectiveness of ZS as a plausible metric for graph diffusion models.

**Strengths:**

1. This paper develops a practical method to capture the topological properties of generated graphs at various resolutions.

2. Theoretical stability guarantees are provided.

3. Extensive experimental results confirm the effectiveness of ZS as a plausible metric for graph diffusion models.

**Weaknesses:**

1. Some illustrations for zigzag persistence and zigzag filtration curves in Section 3 can benefit the convey of the content.

2. Some sentences are not precise, e.g., “M is the number of the” in line 162. Actually, M is a set in the paper.

3. Proposition 3.2 seems to be a simple derivation from proposition 3.2 of Chen et al. (2022). Please further clarify the significance of this proposition. Meanwhile, there exist explicit notation errors in Proposition 3.2 between PDz_alpha_k and PDz’_alpha_k, which should be avoided in the submission.

4. The bootstrap version of ZS is proposed. However, no theoretical analysis of accuracy is provided due to the inherent challenges. I would suggest the authors provide some empirical results or analyses at least.

5. Zigzag spaghetti is proposed as computation efficiency, which remains unclear from a theoretical perspective. Please justify this by comparing its time complexity with existing metrics.

6. Figure 1 is proposed to illustrate the proposed ZS. I suggest the authors provide the necessary explanations for better clarity.

**Questions:**

Please refer to the weakness part.

---

> ### Author Response · Authors · 2024-11-19
> **Rebuttal by Authors (1/2)**
>
> We sincerely thank the reviewer for providing valuable feedback. We detail our response below point by point and we also added the respective updates into the paper (see the parts highlighted in red and Appendix E).
>
> $\textbf{Q1:}$ Some illustrations for zigzag persistence and zigzag filtration curves in Section 3 can benefit the convey of the content.
>
> $\textbf{A:}$ Thank you very much! We agree that this is an important point for conveying the intuition behind zigzag persistence. We have added a new Figure 2 with a toy example on Page 18, Appendix E. We’ll integrate it to the main body of the final version when all other edits/additions/deletions are completed.
>
> From the Figure 2 (c), we observe that although overall there are similarities in curves for all scales, we find that, starting from time 3, the time evolving graphs deliver consistency in terms of topological characteristics of order 1, while there is a notable variability among topological characteristics over the time period from time point 0 to time point 3. Furthermore, another notable topological feature is observed over the time period 1-2, but its lifespan is relatively short.
>
> $\textbf{Q2:}$ Some sentences are not precise, e.g., “M is the number of the” in line 162. Actually, M is a set in the paper.
>
> $\textbf{A:}$ Thank you! Yes, $\mathcal{M}$ is a set containing the observed $p$-dimensional topological features at a given scale. We have updated it in the paper.
>
> $\textbf{Q3:}$ Proposition 3.2 seems to be a simple derivation from proposition 3.2 of Chen et al. (2022). Please further clarify the significance of this proposition. Meanwhile, there exist explicit notation errors in Proposition 3.2 between PDz_alpha_k and PDz’_alpha_k, which should be avoided in the submission.
>
> $\textbf{A:}$ We advance the stability result of Chen et al. (2022) by lifting the key restriction of zigzag curve which considers only a $\textbf{single}$ user-predefined scale $\alpha_{k}$, to a general case, where we show stability for all scales $\alpha_{1}, \alpha_{2}, …, \alpha_{m}$  simultaneously. This involves moving derivations from a univariate zigzag curve in $\mathbb{R}$ to a multivariate zigzag spaghetti in $\mathbb{R}^m$. Practically, this stability result is of high importance to ensure robustness of zigzag spaghetti and the associated graph learning process with respect to uncertainties at $\textbf{all}$ resolution scales $\textbf{simultaneously}$. It also allows us to bypass the subjective selection of a single resolution scale $\alpha_{k}$, which is one of the major criticisms of the zigzag persistence for dynamic data (Xian et al, 2022).
>
> Thanks very much for the careful reading! We have fixed the typos in Proposition 3.2, adding the missing $^{\prime}$.
>
> $\textbf{Q4:}$ The bootstrap version of ZS is proposed. However, no theoretical analysis of accuracy is provided due to the inherent challenges. I would suggest the authors provide some empirical results or analyses at least.
>
> $\textbf{A:}$ In the main body (Table 4 on Page 9), we provided the performance comparison under different bootstrap sample sizes on three graph datasets. We have also conducted additional experiments on COX2 data with bootstrap sample sizes $B \in {10, 20, 50, 100}$ (see Table below).
>
> Similarly to the earlier obtained results (Table 4 on Page 9) and on par with the premise of the resampling methods in the inferential statistics (Efron and Tibshirani, 1993; Hall, 2013), we find that that as the bootstrap sample size increases, the performance of ZS-DM improves and the standard error decreases. This suggests that there likely shall be some theoretical result on asymptotics of bootstrap estimates for ZS-DM, but this is a fundamental question at the intersection of mathematical statistics and DL which we leave for future research.
>
> ++++++++++++++++++++++++++++++++++++++++++++++++++++++
>
> \# Sim $\hspace{0.2ex}$ | 10  $\hspace{8.5ex}$ | 20 $\hspace{8.2ex}$ | 50 $\hspace{8.2ex}$ | 100
>
> ++++++++++++++++++++++++++++++++++++++++++++++++++++++
>
> ZS-DM |  80.35$\pm$1.10 | 80.39$\pm$0.95 | 81.20$\pm$0.90 | 81.73$\pm$0.86
>
> ++++++++++++++++++++++++++++++++++++++++++++++++++++++

---

> ### Author Response · Authors · 2024-11-19
> **Rebuttal by Authors (2/2)**
>
> $\textbf{Q5:}$ Zigzag spaghetti is proposed as computation efficiency, which remains unclear from a theoretical perspective. Please justify this by comparing its time complexity with existing metrics.
>
> $\textbf{A:}$ Zigzag spaghetti has two main advantages (not one): first, it allows for capturing essential topological characteristics of a dynamic object over time at $\textbf{all resolution scales simultaneously}$, and, second, it is easily tractable and computationally efficient (relative to other time-aware topological summaries). To justify the second point, we present the running time comparison among ZS, ZPI, and ZFC for graph classification on MUTAG. The obtained results suggest that ZS-DM achieves the best performance and also delivers competitive running time (in seconds), relative to other zigzag summaries.
>
> ++++++++++++++++++++++++++++++++++++++++++++++++++++++++++++++++++
>
> Data $\hspace{2ex}$ | ZS-DM (ours) $\hspace{8ex}$ | ZPI-DM $\hspace{13ex}$ | ZFC-DM
>
> ++++++++++++++++++++++++++++++++++++++++++++++++++++++++++++++++++
>
> MUTAG | 0.21 sec (91.68$\pm$0.34) | 0.37 sec (90.52$\pm$0.63) | 0.18 sec (90.73$\pm$0.59)
>
> ++++++++++++++++++++++++++++++++++++++++++++++++++++++++++++++++++
>
> In addition, the complexity of zigzag persistence on graphs can be improved for a subclass of graphs satisfying certain assumptions on the size of the unions in the filtration, i.e. of the size of the union of all graphs in the filtration is $\Omega{m^\epsilon}$ for any fixed $\epsilon$, i.e., $0< \epsilon<1$ [1]. However, this is not true for a general class of graphs, and some other more drastic approaches are needed to improve scalability of zigzag persistence on denser graphs. One such potential direction is to compute zigzag persistence based only on landmarks rather than on all nodes and then to use Dowker or witness complexes. This approach has not been yet investigated in computational topology and its theoretical guarantees are yet unknown. We believe though that it’s a very promising approach that can fundamentally shift the scalability problem for ZP.
>
> [1] Dey, T.K. and Hou, T., 2023. Computing Zigzag Vineyard Efficiently Including Expansions and Contractions. arXiv preprint arXiv:2307.07462.
>
> $\textbf{Q6:}$ Figure 1 is proposed to illustrate the proposed ZS. I suggest the authors provide the necessary explanations for better clarity.
>
> $\textbf{A:}$ For Figure 1, we showcase the forward process of the zigzag spaghetti-aware diffusion model. In more detail, at each time step, we first add noise to the data, i.e., transitioning from ${\bf X}_0$ to ${\bf X}_1$ to ${\bf X}_2$ and so on (in this toy example, we suppose there are overall 5 time steps). After that, we use the mixed-up graph construction (MGC) approach to generate the corresponding new mixed-up graph, and then apply the zigzag spaghetti method over the mixed-up graphs within a specific sliding window (i.e., the dashed gray box) to obtain the corresponding zigzag spaghetti.

---

> > ### Comment · Reviewer_HF5K · 2024-11-26
> >
> > Thanks for the detailed responses. Most of my concerns are addressed. However, regarding Q3, this generation from single scale to all scales seems a direct result of the conclusion of Chen et al. (2022). Also in the provided experimental results in Q5, the proposed method ZS-DM does not demonstrate an efficiency advantage over baseline.

---

> ### Author Response · Authors · 2024-11-25
> **A Further Kind Reminder to Reviewer HF5K**
>
> Thanks very much for your time in reviewing and insightful comments, Reviewer HF5K! We sincerely understand you are busy. Since the discussion due is approaching, would you mind checking our response to confirm whether you have any further questions?
>
> We are looking forward to your reply and happy to answer your further questions.

---

> ### Author Response · Authors · 2024-11-27
> **Rebuttal by Authors**
>
> Thanks very much for the helpful feedback! Below our answers on the proposition and computational complexity.
>
> $\textbf{Q7}$: Regarding Q3, this generation from single scale to all scales seems a direct result of the conclusion of Chen et al. (2022).
>
> $\textbf{A}$: On the proposition comment, we respectfully disagree with it. The proposition in Chen et al. (2022) considers a univariate case, i.e. $\mathbb{R}$, and we prove the stability for all cells simultaneously, i.e., for a multivariate case of  $\mathbb{R}^m$. While clearly our result relies on the univariate result of Chen et al. (2022), the proper mathematical derivation of stability for a multivariate case does not exist, and our proposition offers a formal treatment of the multivariate case under the corresponding norms. The step from a univariate case to a multivariate case, while not highly mathematically involved, requires proper mathematical apparatus from linear algebra and operator theory. Of course, we don't argue in any way that the proof is a breakthrough in pure mathematics but it is important to obtain such formal result, ensuring the stability guarantees of a multivariate case in practice.
>
> $\textbf{Q8}$: In the provided experimental results in Q5, the proposed method ZS-DM does not demonstrate an efficiency advantage over baseline.
>
> $\textbf{A}$: On computational efficiency vs. performance, to highlight our point, we have added another experiment on PTC\_FM. We describe the conclusions based on MUTAG (presented earlier) and PTC\_FM (the new experiment) below. The conclusions on the MUTAG and PTC\_FM experiments are consistent.
>
> Table 1: Computational efficiency and performance comparison on MUTAG.
>
> ++++++++++++++++++++++++++++++++++++++++++++++++++++++++++++++++++
>
> Data $\hspace{2ex}$ | ZS-DM (ours) $\hspace{8ex}$ | ZPI-DM $\hspace{13ex}$ | ZFC-DM
>
> ++++++++++++++++++++++++++++++++++++++++++++++++++++++++++++++++++
>
> MUTAG | 0.21 sec (91.68$\pm$0.34) | 0.37 sec (90.52$\pm$0.63) | 0.18 sec (90.73$\pm$0.59)
>
> ++++++++++++++++++++++++++++++++++++++++++++++++++++++++++++++++++
>
>
> Table 2: Computational efficiency and performance comparison on PTC\_FM.
>
> ++++++++++++++++++++++++++++++++++++++++++++++++++++++++++++++++++
>
> Data $\hspace{2ex}$ | ZS-DM (ours) $\hspace{8ex}$ | ZPI-DM $\hspace{13ex}$ | ZFC-DM
>
> ++++++++++++++++++++++++++++++++++++++++++++++++++++++++++++++++++
>
> PTC\_FM | 0.18 sec (66.76$\pm$0.24) | 0.25 sec (62.47$\pm$0.39) | 0.16 sec (61.86$\pm$0.40)
>
> ++++++++++++++++++++++++++++++++++++++++++++++++++++++++++++++++++
>
>
> On ZS vs. ZFC, as ZS considers a multivariate case of filtration scales and ZFC considers only a univariate case of a single filtration scale, ZS and ZFC show overall similar costs, with ZFC costs being, as it can be expected, somewhat lower. However, ZS outperforms ZFC in accuracy (with gains of 7.92\%) and also $\textbf{substantially}$ reduces variability, with gain in std of up to 42\%.
> This phenomenon is not surprising as
> as ZS considers $\textbf{all}$ filtration scales simultaneously.
>
> Now ZS vs. ZPI, ZS outperforms ZPI in computational efficiency up to 43\%, in accuracy up to 6.87\% and in variability up to 46\%.
>
> To summarize:
> 1) in accuracy
> ZS is better than ZFC and ZPI;
> 2) in variability
> ZS is much better than ZFC and ZPI,
> 3) in computational costs,
> ZS is a bit worse than ZFC and is much better than ZPI.
>
>  We hope that these experiments are sufficiently convincing to demonstrate the performance  and computational advantages of ZS. Thanks a lot again for raising these questions which allow us to better highlight the ZS gains.

---

> > ### Comment · Reviewer_HF5K · 2024-11-28
> >
> > Thanks for the additional responses. My concern for Q5 is addressed. Regarding Question 3, I maintain some reservations about my perspective. I appreciate your efforts to formally present a general case derived from a univariate case. However, I perceive the theoretical contribution as somewhat limited according to the current version. Thanks.

---

> > > ### Author Response · Authors · 2024-11-28
> > > **Thankd very much for the feedback!**
> > >
> > > Thanks very much for the feedback, Reviewer HF5K! We are glad that you find your Q5 question is now addressed.
> > >
> > > On Q3, we just would like to reiterate that the theoretical result is new and practically important but we do not claim that its derivation is very mathematically involved or it is a theoretical breakthrough in any sense. We also don't claim that the theoretical result is the primary contribution of the paper. Nevertheless, the stability result in the multivariate case does its job, i.e. it yields certain important garantees in practice.

---

### Official Review · Reviewer_weuX · 2024-11-03

**Soundness:** 3
**Presentation:** 3
**Contribution:** 4
**Rating:** 8
**Confidence:** 3

**Summary:**

This paper introduces zigzag spaghetti (ZS), a novel topological summary designed to enhance diffusion models for graph-based generative AI by capturing higher-order topological properties. Using zigzag persistence, ZS extracts key topological descriptors across multiple resolutions in dynamic graphs, addressing limitations in current models' generalizability. Theoretical stability guarantees for ZS are provided, and it represents the first integration of dynamic topological information in graph diffusion models. Experiments across nine benchmark datasets demonstrate that the proposed method improves performance even up to 10%.

**Strengths:**

1) The proposed method is very interesting and novel. The paper tackles an important topic introducing concepts from computational topology to generative diffusion models for graphs. The motivation behind the idea is clearly communicated and provided.

2) The paper is well written providing a good introduction to the different concepts adopted in the study.

3) The experimental set-up is very strong including multiple baselines with the method outperforming all of the them across all datasets. Furthermore, there are experiments validating the robustness of the method and the computational cost. The paper is very complete on the experimental set-up

**Weaknesses:**

1) The proposed method is not scalable which is rather a limitation. Zigzag persistence introduces additional complexity due to the need for forward and backward inclusions in the filtration sequence, which can become costly for large or dense graphs.

**Questions:**

- I would like to ask the authors if recent efforts in scaling graph diffusion models could benefit also their approach as well. For instance, could techniques such as sparse attention or hierarchical modeling, commonly used in recent large-scale graph diffusion models, be adapted to improve the scalability of the zigzag spaghetti approach?

---

> ### Author Response · Authors · 2024-11-19
> **Rebuttal by Authors**
>
> Thank you for the in-depth review and your encouraging comments! We have addressed your questions below and also added the respective updates into the paper (see the Appendix E).
>
> $\textbf{Q1:}$ The proposed method is not scalable which is rather a limitation. Zigzag persistence introduces additional complexity due to the need for forward and backward inclusions in the filtration sequence, which can become costly for large or dense graphs.
>
> $\textbf{A:}$ Yes, of course, we agree,  in general, zigzag persistence is not computationally cheap. However,  zigzag spaghetti has two main advantages: first, it allows for capturing essential topological characteristics of a dynamic object over time at $\textbf{all resolution scales simultaneously}$, and, second, it is easily tractable and computationally efficient (relative to other time-aware topological summaries). To justify the second point, we present the running time comparison among ZS, ZPI, and ZFC for graph classification on MUTAG. The obtained results suggest that ZS-DM achieves the best performance and also delivers competitive running time (in seconds):
>
> ++++++++++++++++++++++++++++++++++++++++++++++++++++++++++++++++++
>
> Data $\hspace{2ex}$ | ZS-DM (ours) $\hspace{8ex}$ | ZPI-DM $\hspace{13ex}$ | ZFC-DM
>
> ++++++++++++++++++++++++++++++++++++++++++++++++++++++++++++++++++
>
> MUTAG | 0.21 sec (91.68$\pm$0.34) | 0.37 sec (90.52$\pm$0.63) | 0.18 sec (90.73$\pm$0.59)
>
> ++++++++++++++++++++++++++++++++++++++++++++++++++++++++++++++++++
>
> In addition, the complexity of zigzag persistence on graphs can be improved for a subclass of graphs satisfying certain assumptions on the size of the unions in the filtration, i.e. of the size of the union of all graphs in the filtration is $\Omega{m^\epsilon}$ for any fixed $\epsilon$, i.e., $0< \epsilon<1$ [1]. However, this is not true for a general class of graphs, and some other more drastic approaches are needed to improve scalability of zigzag persistence on denser graphs. One such potential direction is to compute zigzag persistence based only on landmarks rather than on all nodes and then to use Dowker or witness complexes. This approach has not been yet investigated in computational topology and its theoretical guarantees are yet unknown. We believe though that it’s a very promising approach that can fundamentally shift the scalability problem for ZP.
>
> [1] Dey, T.K. and Hou, T., 2023. Computing Zigzag Vineyard Efficiently Including Expansions and Contractions. arXiv preprint arXiv:2307.07462.
>
> $\textbf{Q2:}$ I would like to ask the authors if recent efforts in scaling graph diffusion models could benefit also their approach as well. For instance, could techniques such as sparse attention or hierarchical modeling, commonly used in recent large-scale graph diffusion models, be adapted to improve the scalability of the zigzag spaghetti approach?
>
> $\textbf{A:}$ We greatly appreciate this reviewer’s suggestion! Based on the paper [1], we incorporate its sparse denoising network (which contains a graph transformer architecture with a sparse attention mechanism) into our zigzag spaghetti approach and have conducted additional experiments on MUTAG and BZR datasets. We call the zigzag spaghetti model with sparse denoising network ZS-DM$_{new}$.
>
> From the Table below, we observe that ZS-DM$_{new}$ always outperforms the previous version (i.e., ZS-DM$\_{old}$) on both MUTAG and BZR datasets. Thanks very much for this excellent suggestion for further improvement of the ZS approach! We’ve added it to the paper, Appendix E (Page 18) at this point, and we’ll integrate it to the main body of the final version when all other edits/additions/deletions are completed.
>
> ++++++++++++++++++++++++++++++++++++++++++++++
>
> Model $\hspace{2.2ex}$ | MUTAG $\hspace{3.8ex}$ | BZR
>
> ++++++++++++++++++++++++++++++++++++++++++++++
>
> ZS-DM$_{old}$ $\hspace{0.2ex}$ | 91.68$\pm$0.34 | 86.20$\pm$0.12
>
> ++++++++++++++++++++++++++++++++++++++++++++++
>
> ZS-DM$_{new}$ |  92.15$\pm$0.32 | 86.39$\pm$0.10
>
> ++++++++++++++++++++++++++++++++++++++++++++++
>
> [1] Qin, Y., Vignac, C. and Frossard, P., 2023. Sparse training of discrete diffusion models for graph generation. arXiv preprint arXiv:2311.02142.

---

> > ### Comment · Reviewer_weuX · 2024-11-19
> > **Answer to the authors**
> >
> > I would like to thank the authors and acknowledge the rebuttal answers provided. The authors have presented additional experiments and a more scalable version of their proposed model, which appears very promising. As mentioned in their response, they plan to integrate these improvements into the main body of the paper. Given this, I will maintain my current score.

---

> > > ### Author Response · Authors · 2024-11-19
> > > **Thanks so much!**
> > >
> > > We are very grateful for the constructive and motivating response and for appreciating the potential of our approach! The new experiments are already in the paper (Appendix for now) but we'll bring them to the main body.

---

### Meta-Review · Area_Chair_BSqS · 2024-12-21

**Metareview:**

The authors propose Zigzag Spaghetti (ZS), a time-aware topological summary designed to capture joint higher-order shape properties across a sequence of graphs at multiple resolution scales. The core ide involves leveraging zigzag persistence in a computationally efficient manner. The authors prove the theoretical stability of ZS and demonstrate its practical utility in diffusion-based prediction and classification tasks on graphs.

The reviewers largely agreed with the novelty of the proposed method, its theoretical stability, strong empirical effectiveness, and clear presentation. The paper offers sufficient merit for acceptance.

**Additional Comments On Reviewer Discussion:**

Most concerns were addressed during the discussion period, except for one reviewer noting that the theoretical contribution, while important, is relatively straightforward. This point, which the authors also acknowledged, seems minor in light of the other contributions of the paper.

---

### Decision · Program_Chairs · 2025-01-22

Accept (Poster)